# Conformational Heterogeneity and Frustration of the Tumor Suppressor p53 as Tuned by Punctual Mutations

**DOI:** 10.3390/ijms232012636

**Published:** 2022-10-20

**Authors:** Anna Rita Bizzarri

**Affiliations:** Biophysics and Nanoscience Centre, DEB, Università della Tuscia, Largo dell’Università, 01100 Viterbo, Italy; bizzarri@unitus.it; Tel.: +39-0761-357031

**Keywords:** p53, mutated p53, molecular dynamics simulations, collective modes, energy landscape, frustration

## Abstract

The conformational heterogeneity of the p53 tumor suppressor, the wild-type (p53wt) and mutated forms, was investigated by a computational approach, including the modeling and all atoms of the molecular dynamics (MD) simulations. Four different punctual mutations (p53R175H, p53R248Q, p53R273H, and p53R282W) which are known to affect the DNA binding and belong to the most frequent hot-spot mutations in human cancers, were taken into consideration. The MD trajectories of the wild-type and mutated p53 forms were analyzed by essential dynamics to extract the relevant collective motions and by the frustration method to evaluate the degeneracy of the energy landscape. We found that p53 is characterized by wide collective motions and its energy landscape exhibits a rather high frustration level, especially in the regions involved in the binding to physiological ligands. Punctual mutations give rise to a modulation of both the collective motions and the frustration of p53, with different effects depending on the mutation. The regions of p53wt and of the mutated forms characterized by a high frustration level are also largely involved in the collective motions. Such a correlation is discussed also in connection with the intrinsic disordered character of p53 and with its central functional role.

## 1. Introduction

p53, often called the tumor suppressor protein, plays a crucial role in several biological functions, such as cell cycle progression, apoptosis induction, and DNA repair [1]. p53 is a homo-tetrameric protein, with each monomer consisting of an N-terminal transactivation domain (NTD), a C-terminal domain (CTD), and a large-structured region core domain (DBD) which binds the DNA response elements [2,3,4,5]. The p53 activity is finely tuned by different processes, such as the interaction with some molecules, such as MDM2, S100, COP1, and, interestingly, miRNA [6,7]. When p53 does not work properly, for example as a consequence of a mutation, a drastic increase in diseases, such as cancers, may occur [8]. In more than half of human cancers, mutations are located in the DBD portion [9,10,11]; this can strongly alter the capability of p53 to perform its tumor suppressive functions and sometimes leads p53 to acquire new functions (gain-of-function, GOF), including transcriptional activation of different genes [12]. P53 is characterized by the presence of some structurally disordered regions which prevent the biomolecule from a full 3D characterization [13]. Accordingly, p53 belongs to the class of intrinsically disordered proteins (IDPs) whose partial structural disorder gives rise to the co-existence in the solution of slightly different conformations [14]. Such a heterogeneity confers to IDP systems the capability of performing different functions, often through the binding to various biological molecules [15,16].

From a physical point of view, IDPs can be described within the Frauenfelder’s framework which represents the protein energy landscape as being characterized by several nearly isoenergetic local minima; with such a picture being reminiscent of that describing other complex systems, such as glasses, spin glasses, etc. [17,18]. Briefly, competing interactions among components do not allow these systems to reach a global minimum, instead a plethora of local minima emerge and are explored during the functional dynamics [19]. Additionally, it is assumed that protein molecules have evolved following the principle of minimal degeneracy, or frustration, for which the energy of a protein progressively decreases as far as it approaches its native state [19]; some residual frustration are still present to facilitate the fundamental biological functions [20]. In this framework, IDPs constitute a particular class of proteins whose partial disorder introduces a further contribution to the frustration of the energy landscape [21]. For its prominent biological role, p53 is an extremely intriguing example of IDPs whose understanding deserves a large interest from both the biological and physical point of view.

Very recently, we have put into evidence, by time resolved fluorescence combined with MD simulations, a conformational heterogeneity of the DBD portion of p53 in the nanosecond temporal window [22], with such a heterogeneity undergoing a significant modulation upon varying the external conditions. Here, we are interested to closely address such a conformational heterogeneity in the same temporal window and in the whole p53 molecule. On the other hand, motions within the time interval of hundreds of nanoseconds are relevant for many biological processes [23].

With such an aim, we have applied a computational approach to wild-type p53 (p53wt) and four mutated p53 forms, with each one bearing a punctual mutation. At this stage, the analysis has been restricted to a p53 monomer. Preliminarily, a model for the whole structure of the p53wt and of the four mutants have been derived by combining the available information from the Protein Data Bank (PDB), with previously developed approaches [24]. All these p53 systems have been submitted to molecular dynamics (MD) simulations for a relaxation, followed by 100 ns long runs for data analysis. The obtained data have been analyzed by two methods: essential dynamics (ED) and frustration analysis. The ED method, based on principal component analysis (PCA), allows us to extract collective modes from the conformational space explored by a biomolecule during the MD runs [25]; the collective mode can be described as the occurrence of simultaneous large amplitude motions involving different parts of a biomolecule. Upon the assumption that the major collective modes dominate the functional dynamics, ED has been widely used to extract motions which could be relevant for the protein functionality. Furthermore, frustration analysis, based on an energy landscape-inspired theory, allows us to investigate the energetic degeneracy of protein molecules. Accordingly, protein sites with a high local frustration are expected to correlate with functional regions related to both binding sites and the regions involved in allosteric transitions [26].

We have analyzed wild-type p53 (p53wt), and four punctual mutations (p53R175H, p53R248Q, p53R273H, and p53R282W) which are part of mutations that affect DNA binding and belong to the most frequent hot-spot mutations in human cancers [27,28,29,30]. P53R175H, in which Arg175 is replaced by His, is one of the most common p53 mutants found at progressively higher quantities in several tumors, such as lung, colon, and rectal tumors, and with a particularly high incidence in breast tumors [31,32,33]. This mutation abolishes the p53wt ability to recognize its target genes which are involved in tumor suppression, yet promotes new GOF functions, through direct or indirect associations with DNA [31,34]. P53R248Q, in which Arg248 is replaced by Gln, is one of the most common mutants found in breast cancers and its presence is also largely used as a patient death predictor [35]. This punctual mutation yields a weakness of the binding to DNA with a concomitant increase in aggregation propensity [36]. The p53R273H mutant, in which Arg273 is replaced by His, is characterized by a reduction in the DNA binding capability and it is associated with an inhibition of the autophagy machinery, promoting cancer cell survival [37]. Finally, the p53R282W, in which Arg282 is replaced by Trp, is associated with a shorter survival time. This mutant still retains a partial transcriptional ability, however it exhibits a significant aggregation propensity [36,38]. All the results have been discussed in connection with the available information about the effects as induced by the punctual mutations.

## 2. Results and Discussion

### 2.1. Analysis of General Properties

Figure 1 shows the best model for the structure of the p53wt monomer (see Section 3.1) with the four amino acids submitted to a mutation, represented as colored spheres. All these residues are located in the northern part of the molecules and they are clustering around the Zn ion, which is involved at the DNA binding region.

As described in Section 3.2, each system has been submitted to a preliminary run to allow for a relaxation of the structures. Successively, 150 ns long runs have been carried out. Figure 1B shows the representative temporal evolution curves of all the atoms of RMSD, during the last 150 ns long MD trajectories. In all the cases, we note a rather rapid increase within the first 5 ns, followed by a slower increase which is practically completed within the first 50 ns. Then, the RMSDs oscillate around a plateau, consistently with the establishment of a stable state. The observed trend is qualitatively similar to that usually detected in the RMSD of other proteins, although the time required to reach a plateau is generally longer.

Such an effect together with the rather high RMSD values reached at the end of the run (ranging from 0.4 to 0.8 nm) find a correspondence with the presence of some disordered regions which confer to the protein molecule a higher flexibility. The last 100 ns of the runs have been used for the data collection.

Table 1 reports the average and the corresponding standard deviation, as derived from all the 100 ns long runs, for a collection of properties (the root mean square displacement (RMSD), surface accessible surface area (SASA), and gyration ratio (Rg)) of the five p53 systems. The rather high RMSD values (ranging from 0.4 to 0.8 nm) are consistent with the presence of some disordered regions which confer to the protein molecule some flexibility. P53R248H shows a higher RMSD value with respect to that of p53wt, while the other mutants are characterized by similar values to that of p53wt. Additionally, both the gyration radius, Rg, and the SASA values of all the mutated forms are almost the same as those of p53wt, with this indicating that the global dimension of p53 is not affected by the punctual mutations. Furthermore, the fraction of the amino acid residues involved into the secondary structure, averaged from the last 10 ns of the runs and reported in the last column of Table 1, does not exhibit significant changes in comparison to that of p53, indicating that, in the analyzed temporal window, mutations do not induce drastic changes in the p53 structure.

Figure 2 shows the representative RMSD and RMSF as a function of the residue number, for all the systems; the position along the chain corresponding to the mutated residues is marked by black stars, while the grey squares indicate the positions of the main amino acids directly involved in the DNA binding. The largest part of the peaks in both the RMSD and RMSF is located at almost the same position in the various systems, although they exhibit a large variability in the peak intensity, similarly to what was observed in ref. [24]. In some cases, the deviation of the RMSD and RMSF of mutants with respect to those of p53wt, exceeds the run-to-run variability. In particular, the RMSDs of p53R248Q show high deviations around the 32, 320, and 379 residues, while for 53R175H, higher deviations occur at the 186, 289, and 302 residues. Notably, no significant changes are detected around the mutations. However, we remark that structural information about these mutants and their changes, with respect to p53wt, is rather poor (see, e.g., refs. [39,40]); the use of techniques such as small-angle X-ray scattering and kinetic hydrogen-deuterium exchange mass spectroscopy could help to improve the knowledge of the structural properties of p53 mutants [41,42].

Concerning RMSF, higher values than those of p53wt appear around 312–316 residues for both p53R273H and p53R248Q, while lower values are observed at the residue 85 for p53R282W. In summary, in the analyzed temporal window, the general properties of p53 mutated forms do not significantly deviate from those of p53wt.

### 2.2. Analysis of p53 Collective Modes

To find out the relevant collective modes of wild-type and mutated p53 molecules, the MD trajectories have been investigated by the ED method. Indeed, ED allows us to extract concerted fluctuations with large amplitudes from MD simulations, through PCA. In particular, collective modes can be described by the eigenvectors derived from the covariance matrix of atomic fluctuations, with the most relevant modes corresponding to the highest eigenvalues. The first 30 eigenvalues, ranked in a decremental order, are shown in Figure 3A. In all the cases, the eigenvalues rapidly decrease down to a very low value (less than 5% of the maximum) within the first ten eigenvalues. Similar results have been obtained for other runs. Such a trend, commonly observed in the eigenvalues of protein modes, indicates that the most relevant motions are concentrated in a few modes whose dimensionality is represented by the number of eigenvalues accounting for the first ranked values.

Table 2 reports the average percentages of the ratio between the sum of the first ten eigenvalues over the sum of all the ones (column 2). The first ten eigenvalues capture more than 60% of the total in all the cases, confirming that the dynamics of p53 are dominated by a few principal modes. No significant changes among the various systems are observed. This indicates that the investigated mutations do not drastically affect the global dynamics of p53 consistently with the capability of these mutants to perform, at least in part, their physiological role.

Based on these results, global information about the configurational space sampled by the p53 systems, can be derived from the first ranked eigenvectors. Representative 2D-plots of the projection on the first eigenvector (PC1) vs. the second one (PC2) are collectively shown in Figure 3B, using colored spots. In all the cases, the spots cover a wide region, indicating the establishment of wide collective modes in both wild-type and mutated p53 molecules. The covered width of PC1 and PC2, averaged over all the runs, are also reported in Table 2. Notably, for p53wt, the width values are higher than those previously found for the DBD portion of p53 (of about 15 nm) [43]. Accordingly, wider collective motions are established in p53wt with respect to those detected for its DBD portion in the same temporal window. This can be put into a relationship to the presence in the whole p53 molecule of the CTD and NTD regions (missing in the DBD), which confer to the molecule a higher flexibility. Indeed, these regions, being devoted to interacting with several different ligands, are expected to require some heterogeneity for performing this “hub-like” function [1,13].

PC1 and PC2 of p53R248H and p53R273H cover a wider region in comparison with p53wt, while they have almost the same extensions for p53R175H and p53R282W, indicating that punctual mutations can modulate collective modes of p53. Under the assumption that collective motions play some role in the functionality, these changes are expected to have some impact on biological response, e.g., on the capability of p53 to interact with ligands.

To closely address the effects of punctual mutations on collective motions in the various part of the p53 molecule, we have analyzed the extreme projections on the eigenvectors. The representative snapshots of the extreme projections on the first two eigenvectors of p53wt are shown in Figure 4. In both the cases, we note the presence of collective modes throughout the whole protein structure, whose extension, however, varies from region to region. Regions of p53wt characterized by displacements between the two extreme projections higher than 0.6 nm are marked by orange sticks. As expected, more diffused and wider displacements are detected for the first eigenvector with respect to the second one. The largest displacements appear within both the random coils and α-helices located at the CTD and NCT portions. Since these parts of p53 are largely involved in the binding to different partners, such a result is consistent with a functional role of the collective motions. Additionally, the rather restricted collective modes in the regions involved in β-sheets can be ascribed to the fact that β-sheets provide the skeleton of the molecule, whose structure should be preserved for a higher stability of the molecule.

The region close to the Zn ion, which is involved in the DNA binding, is also characterized by small collective motions. It could be speculated that the presence of restricted modes in this part could be related to an optimized interaction of p53 with the DNA.

Figure 5 shows the representative snapshots of the extreme projections on the first eigenvector for the four mutated p53 molecules; again, the differences between the two extreme projections of the eigenvectors higher than 0.6 nm are marked by red sticks. These results find a correspondence with the global properties, as reported in Table 2. Similarly to what was observed for p53wt, rather wide collective modes are detected at both the random coils and α-helices, while the β-sheets are practically not involved. At visual inspection, the amplitude of the collective modes appears wider for p53R248Q (Figure 5B) than those of the other mutants, consistently with what was observed in the 2D-plot. In particular, this mutant exhibits wide collective motions at both the CTD and the NTD regions, with such a result finding a correspondence with the high propensity of these regions to form amyloid fibrils, as recently demonstrated [36]. Indeed, wide collective motions could enlarge the available configurational states, and then it could increase the interaction capability with additional molecular species. In this respect, it has been recently observed that changes in the mutation may allosterically propagate to the CTD and NTD regions, from which fibril nucleation is believed to start [36].

Both the p53R175H and p53R282W mutants (Figure 5A,D) show reduced collective modes which, however, involve the whole structure, which is consistent with the data reported in Table 2. Such a behavior is in agreement with the observed global effects on the p53 molecule as a consequence of these two punctual mutations. Indeed, the two mutations in p53R175H and p53R282W have been found to cause global conformational distortions which, in turn, affect the binding of the DNA. On the other hand, the other two mutants (p53R248Q and p53R273H) are classified as contact mutants since their mutations mainly affect the contact with the DNA [28,29,30]. Interestingly, the H_2_-helix, which directly interacts with DNA, is involved in collective motions in all the mutants. Such a behavior is consistent with a modulation of the capability of these mutants to interact with DNA as experimentally observed [44]. Finally, we note that the region around the punctual mutations shows some wider collective modes for both the p53R248Q and p53R273H mutants. Generally, the establishment of new collective motions, or their enhancement in mutants, with respect to p53wt, may engender new binding capabilities and then new functionalities, in connection with the GOF hypothesis.

Therefore, the ED analysis on wild-type and mutant p53 molecules indicates the presence of wide collective modes through the molecule, with a large involvement of both random coils and α-helices in regions where the binding of ligands is expected; this may have some effect on the p53 response. These results support the conformational heterogeneity in the analyzed temporal window, that punctual mutations can induce some change in the collective modes with possible consequences on the p53 functionality.

### 2.3. Analysis Frustration

The degeneracy of the energy landscape, called also frustration, in wild-type and mutated p53 molecules has been investigated by applying the frustration analysis, as described in Section 3.3. We have first evaluated the total frustration index, which is a parameter developed within the network theory to quantify the degeneracy of a local minima in the energy landscape and widely used in different fields, such as physical, chemistry, biological, and social systems [45]. In proteins, the frustration index is related to the contacts between amino acids and it quantifies how much a residue pair contributes to the energy of a given structure compared to what it would contribute in a typical molten globule configuration [26]. The total frustration index, obtained by averaging over all the residues and all the runs from the last 10 ns of the run, is reported in Table 2. For p53wt, we found a frustration index of 7.3, while a higher value is detected for the p53R248Q. The other mutants are characterized by a total frustration index, which is almost the same of that of p53wt. Since the local properties of the structural and dynamical properties of a protein molecule are crucial for its functional response, we have investigated the local frustration around each residue of the p53 molecules. In particular, we have analyzed the snapshots and the related frustration maps of the average structures as derived from the last 10 ns of the run. A representative snapshot and the related map of p53wt are shown in Figure 6 and Figure 7, respectively. In the map, red and green spots indicate residues with a high and low frustration, respectively, with spots along the diagonal providing a connection between residues which are close along the chain, while spots out of the diagonal indicate spatially close residues, but they belong to different parts of the molecule. In snapshots, connections between residues characterized by a high frustration are marked by red dashed lines, while those with a low frustration are marked by green dashed lines. In the following, we have focused our attention to the clusters of spots, by labeling with C the main clusters along the diagonal and with O those out of the diagonal.

In the map shown in Figure 6, several spots can be visualized. Red spots, corresponding to a high frustration, are observed along and out of the diagonal. Three main clusters of red spots, labeled as C1, C2, and C3 and marked by blue circles, appear along the diagonal. Some red spots can be also observed out of the diagonal, with two small clusters, labeled as O1 and O2. The regions corresponding to a high frustration can be better visualized in the related snapshot, shown in Figure 7. Clusters C1 and C3 involve the random coils of the NTD and CTD, respectively, while C2 is at the H_1_-helix. The two small clusters, O1 and O2 connect the NTD and CTD of the molecule. Furthermore, the region close to the Zn ion is also characterized by the presence of some frustration. We are also interested to discuss the results from the frustration with those obtained by ED analysis. A comparison between Figure 4A and Figure 7 suggests some correlation between the frustration level and the collective motions. In particular, regions characterized by wide collective motions are also characterized by a high frustration level.

Figure 8 shows a representative snapshot of the p53R175H mutant. Regions of the molecule where the main clusters have been detected in the map are again marked by C and O letters. At the H_1_-helix, the cluster C2, shared with the p53wt, is detected. New clusters (labeled as C4-C6 and O3) appear at different positions. The C4 cluster, partially involving residues located in the back part, also involves a random coil of the NTD. Additionally, C5 occurs at the DBD and C6 at the CTD; the O3 cluster involves residues belonging to a different portion of the random coil in the NTD. Notably, at the CTD region, where the C3 cluster has been detected in p53wt, no frustration appears. Since this portion is involved in the binding of several ligands, the loss of frustration could be related to a decrease in the p53R175H functionality, with respect to p53wt. Again, a comparison of the results obtained for the frustration and collective motion of p53R175H (see Figure 5A) puts into evidence that there is some correlation between the frustration and collective motions; this can be seen, e.g., for residues around C2 and C4.

The snapshot of p53R248Q, shown in Figure 9, exhibits a rather high level of frustration throughout the molecule, consistent with the high value for its total frustration index. Three main clusters (labeled as C2, C3, and C4), involve residues close along the chain. C2 and C3 are shared with p53wt, while C4 is shared with p53R175H. The cluster O4, involving residues out of the diagonal, connects different portions in the CTD part. The high frustration level detected at the CTD could be put into relationship to the high propensity of this regions to form fibrils, as recently put into evidence by [36]. Indeed, a high frustration can give rise to several slightly different conformations which, in turn, could increase the possibility of interacting with surrounding molecules. A comparison between Figure 5B and Figure 9 shows that C1 and C2 are characterized by both a high frustration and wide collective motions.

The snapshot of p53R273H, shown in Figure 10, is characterized by a close similarity to that of p53wt. Indeed, we note the presence of the C1, C2, and C3 clusters along the diagonal shared with p53wt, and a further cluster, C4, shared with p53R175H, involving the NTD. Additionally, we note that the O2 cluster connecting the NTD and CTD portions is also shared with p53wt. From a comparison between Figure 5C and Figure 10, it emerges that the regions around C1, C2, and C3 are also characterized by a high frustration and wide collective motions.

Finally, the snapshot of p53R282W, shown in Figure 11, again exhibits the C1, C2, and C3 clusters, shared with p53wt, and a further rather dense cluster C5, shared with p53R175H. Additionally, we note the cluster O5, involving different portions of the DBD and even regions around Zn, exhibits a high frustration; such results deserve some interest in connection with the DNA binding properties. Furthermore, by comparing Figure 5D and Figure 11, again we note some correlation between the regions’ high frustration and the wide collective motions, such as those around C1, C3, and O5.

Globally, these results show that the p53 monomer is characterized by a high level of frustration, mainly in regions forming random coils, with these regions being also involved in binding with ligands. As already mentioned, IDP molecules, to which p53 belongs, are often required to bind several different ligands to perform their functional role. Accordingly, it can be speculated that the high frustration in these biomolecules could be the result of an evolution to fulfil their functional role. Along this direction, the evidence that punctual mutations can locally alter frustration could provide an explanation for the effects of mutations on the functionality. Furthermore, it is interesting to note that a punctual mutation can modify frustration, even at regions far from the mutations, with this supporting the complex character of this protein, for which even a tiny change in a part of the system could alter its global behavior.

Finally, it deserves some interest to discuss the results about frustration in connection with those of the collective modes. The qualitative comparison between the frustration level and the presence of collective motions in the p53 molecule indicates that collective motions mainly appear in regions characterized by a high frustration, with this suggesting a close interplay between these two aspects. Indeed, a high frustration is related to the presence in the energy landscape of many nearly isoenergetic minima, which are expected to correspond to slightly different local arrangements of protein atoms (i.e., conformational states). On the other hand, a collective mode, which involves various parts of the biomolecule, could be described as a collection of different local conformational states of the biomolecule; with this being likely finalized to perform a specific role. In this picture, a high frustration index of a molecular portion could be a required condition to give rise to collective motions. In other words, the energy landscape of a protein could be evolved to assure the presence of several minima (a high frustration) in order to assure a variety of different conformational states, eventually leading to the establishment of collective modes, which could be essential to perform a given functional role.

## 3. Computational Methods

### 3.1. Modeling of p53 Molecules

The model of the full-length wild-type p53 monomer (393 amino acid residues) was built from the amino acid sequence using the I-TASSER suite [46,47], including the B chain of the DBD portion (94–289 amino acid residues), in complex with a consensus DNA (1TUP entry from the protein data bank) [5]. The available information about the missing portion which includes the N- and C-term was taken into consideration in I-TASSER. The predicted protein structure was characterized by a medium confidence with a C-score of 2.37 and a TM score of 0.44–0.14, for Model 1. According to ref. [24], we assumed that Model 1 provides a realistic starting point for the MD simulation to investigate the intrinsic flexibility of p53. The Zn ion, which is tetrahedrically coordinated to the side chains of Cys176, His179, Cys238, and Cys242, forms a Zn-finger motif connected to the L2 and L3 loops [48]. The interaction of the Zn ion with its ligands was treated through a bonded approach in which the Zn-N and Zn-S bonds and the S-Zn-S angles were set according to the parameters provided in refs. [49,50,51,52]. Punctual mutations were inserted by the Swiss PDB Viewer software starting from the p53wt model [53]. In particular, in p53R175H, Arg175 was replaced by His; in P53R248Q, Arg248 was replaced by Gln; in p53R273H, Arg273 was replaced by His; and finally in p53R282W, Arg282 was replaced by Trp.

### 3.2. Molecular Dynamics (MD) Simulations and Analysis

MD simulations of wild-type and mutated p53 molecules in water were carried out by the GROMACS 2018 package [54], using AMBER03 and SPC/E Force Fields for the protein [55] and for water [56], respectively. Each p53 molecule was centered in a cubic box, with dimension 9.0 × 9.0 × 9.0 nm^3^, filled with water molecules to assure a minimum hydration level of 8 g water/g protein. A number of water molecules, ranging from 21,973 to 22,035, were added to the various protein systems. The ionization states of the protein residues were fixed at pH 7, and 36 water molecules were randomly substituted with an equal number of Na^+^ ions to keep the systems electrically neutral. The final systems had a total number of atoms ranging from 71,941 to 72,151. Simulations were performed by following the same procedures used in refs. [52,57]. Briefly, the Linear Constraint Solver (LINCS) algorithm was utilized to constrain the H bonds [58], while the electrostatic interactions were computed using the particle mesh Ewald (PME) method with a lattice constant of 0.12 nm [59,60]. Each system was submitted to an energy minimization procedure by using the steepest descent method. After minimization, the system was heated to 300 K with steps at 50, 100, 150, and 250 K and submitted to a 50 ns long MD trajectory for equilibration at 300 K, followed by further a 150 ns for the data collection. Periodic boundary conditions in the NPT ensemble with T = 300 K and *p* = 1 bar, with a time steps of 2 fs. The temperature was controlled by the Nosé-Hoover thermostat with a coupling time constant τ_T_ = 0.1 ps [61], while the Parrinello–Rahman extended ensemble, with a time constant τ_P_ = 2.0 ps, was used to control the pressure [62]. For each system (wild-type protein and mutated forms), five independent runs were performed, and the MD trajectories were analyzed by the GROMACS package tools [54]. The temporal evolution of the resulting trajectories was monitored by following the Root Mean Square Displacement (RMSD), the Root Mean Square Fluctuation (RMSF), the Surface Accessible Surface Area (SASA), and the gyration ratio (Rg) according to the methods described in ref. [63]. The figures were created with Pymol and VMD [53,64].

### 3.3. Essential Dynamics (ED) Analysis

The MD trajectories were analyzed by PCA and, in particular, by the ED method implemented in GROMACS [65,66]. Such an approach identifies a new reference frame to describe the overall dynamics of the system, allowing us to extract the protein motions which are mostly contributing to the overall dynamics [67]. ED was carried out by the Covar and Anaeig subroutines of GROMACS [65,66]. After a least square fit to remove the roto-translations, the covariance matrix Cij was calculated by:(1)$$C_{\mathrm{ij}}=\langle (x_{i}-\langle x_{i}\rangle )\left(x_{j}-\langle x_{j}\rangle \right)\rangle$$
where x_i_ are the x, y, and z Cartesian coordinates of the C_α_ atoms of the DBD and the <> brackets indicate an average over the trajectories. C_ij_ was diagonalized by finding out a set of eigenvalues and eigenvectors; with the eigenvectors corresponding to directions in an N dimensional space and are called principal coordinates (PCs), while the eigenvalues represent the total mean square fluctuations along the corresponding eigenvectors.

### 3.4. Frustration Analysis Method

The energy landscape of the p53 molecules was investigated by applying the algorithm implemented in the Protein Frustatometer 2 Server http://frustratometer.qb.fcen.uba.ar/ (accessed on 19 October 2022). The frustration index ($F_{i,j}^{m}$), or Z-score, for the contact between the amino acids i and j was calculated through the following expression [68,69]:(2)$$F_{i,j}^{m}=\frac{E_{i,j}^{N}-\langle E_{i,j}^{U}\rangle }{\sqrt{1/N\sum_{k=1}^{n}{\left(E_{i,j}^{N}-\langle E_{i,j}^{U}\rangle \right)}_{}^{2}}}$$
where $E_{i,j}^{N}$ and $E_{i,j}^{U}$ are the pairwise interaction energies of the molecule in the reference (U) and in the decoy configuration (N), respectively; <$E_{i,j}^{U}$> being the average energy of the decoys. For the analysis, a region of 5 Å from the C_α_-atom for each residue was used. The energy distribution of the decoy was calculated using the configurational frustration, in which the native interaction between the two residues was measured by randomly selecting amino acids from the protein composition and even by changing the location. The energy includes the electrostatics with the k constant fixed at 4.15, which corresponds to a relative electric constant of 80. The frustration index of the p53 structures was calculated by averaging 100 snapshots sampled in the last 10 ns of the 150 ns long runs.

## 4. Conclusions

Our computational analysis allowed us to investigate the collective modes and the degeneracy of the energy landscape of p53wt and the four mutated forms, in the temporal window of 100 ns. The ED method revealed the establishment of collective modes in p53, with a large involvement of the regions containing the binding sites of physiological ligands; this supports a strong connection between the collective modes and the functionality. The amplitude of the collective motions undergoes some changes varying with the specific mutation and differently affecting the various regions of the molecule. The effects, as due to the punctual mutations on the global dynamical behavior of p53, could be related to the acquisition of a new functionality for mutated p53, consistent with the GOF picture. The frustration analysis indicated: (i) regions with a high frustration are the same and largely involved in collective motions and (ii) punctual mutations can significantly modulate frustration, even of regions far from the mutations. These results confirm the intricate relationship between the structure and dynamic of p53, suggesting some impact on its functionality. Although the capability to affect the functionality by acting on the p53 structure (e.g., by introducing punctual mutations) is still far from being reached, these results provide some clues on how a punctual mutation can modulate the energy landscape, and then the functionality.

## Figures and Tables

**Figure 1 ijms-23-12636-f001:**
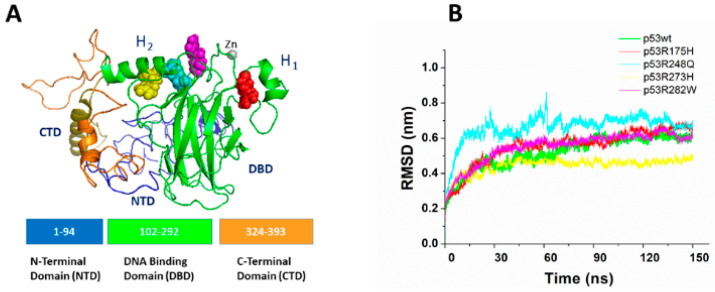
(**A**) Graphical representation of the best model for p53wt monomer. The residues submitted to a punctual mutation are represented as colored spheres: p53R175H (red), p53R248Q (cyan), p53R273H (yellow), and p53R282W (magenta). The Zn ion is represented as a light grey sphere. The various p53 domains are marked in different colors. (**B**) Temporal evolution of the RMSDs for all the p53 systems.

**Figure 2 ijms-23-12636-f002:**
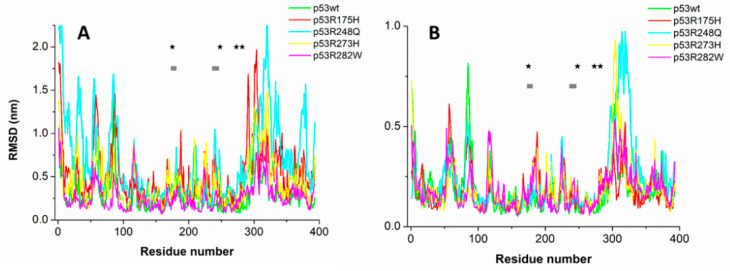
(**A**) RMSD and (**B**) RMSF as a function of the residue number. Positions of the residues submitted to mutations are marked by black stars, while those of residues involved in the DNA binding are marked by grey squares.

**Figure 3 ijms-23-12636-f003:**
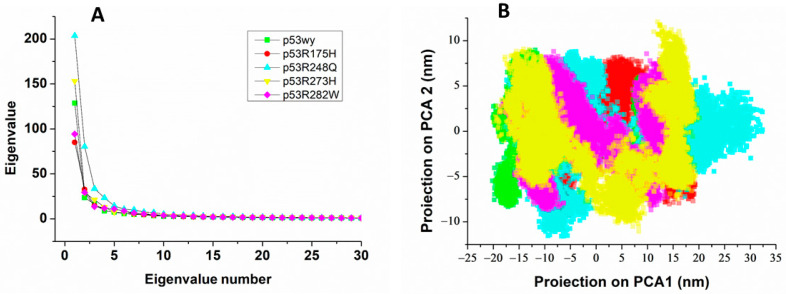
(**A**) Eigenvalues as a function of the eigenvalue number; (**B**) 2D-plot of projection of PC1 vs. the projection of PC2 for all the systems. Colors are the same as in Figure 1.

**Figure 4 ijms-23-12636-f004:**
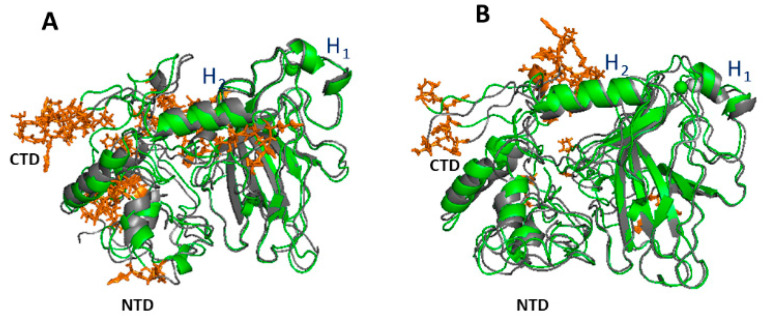
Extreme trajectory conformations projected along: (**A**) the first and (**B**) the second eigenvector for p53wt. The Zn ion is represented as a sphere. Regions characterized by displacements between the two extreme projections higher than 0.6 nm are marked by orange sticks.

**Figure 5 ijms-23-12636-f005:**
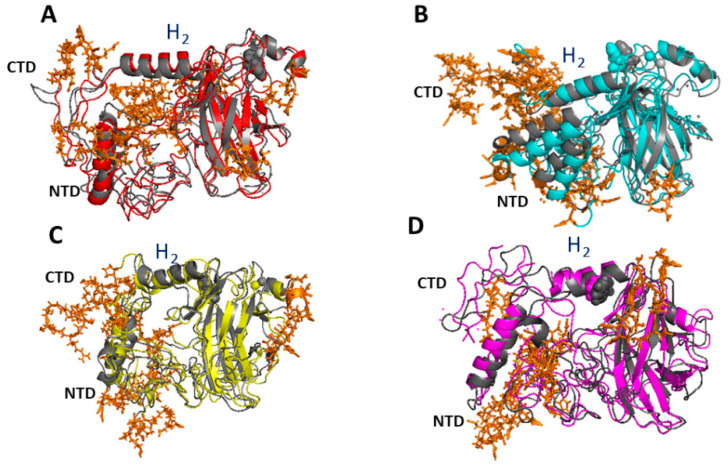
Extreme trajectory conformations projected along the first eigenvector for the four mutated p53 molecule. (**A**) p53R175H; (**B**) p53R248Q; (**C**) p53R273H; and (**D**) p53R282W. Mutated residues and Zn ion are represented as spheres. Regions characterized by displacements between the two extreme projections higher than 0.6 nm are marked by orange sticks.

**Figure 6 ijms-23-12636-f006:**
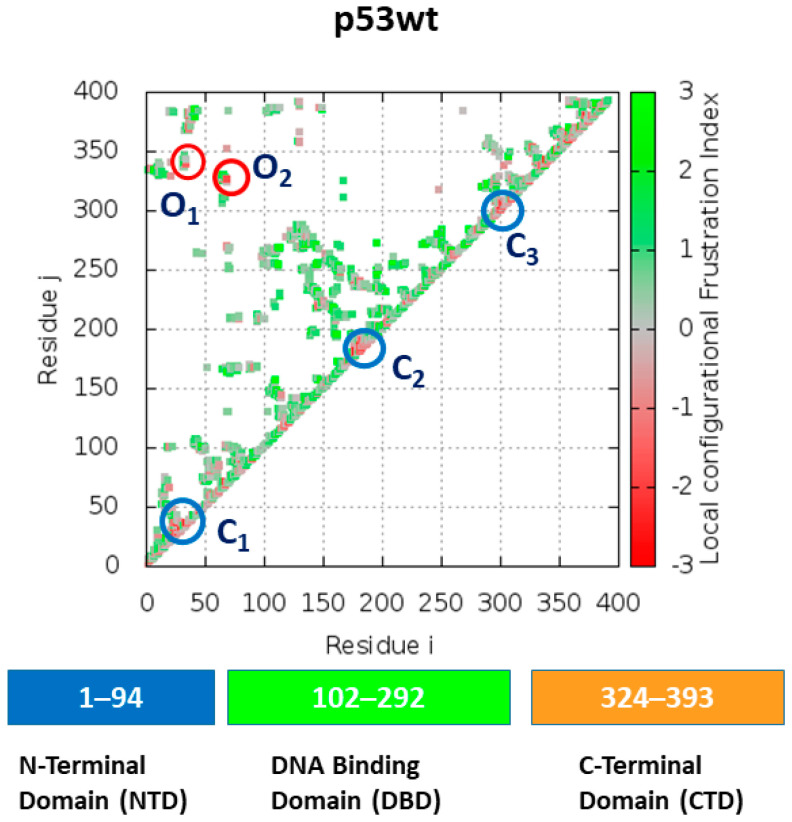
Frustration map of p53wt showing colored spots to indicate the level of frustration: green represents minimal frustration and red high frustration. The main clusters along the diagonal, corresponding to connections between residues close along the chain, are indicated by blue circles. The main clusters out of the diagonal, corresponding to connections between residues spatially close but belonging to different part of the molecule, are indicated by red circles.

**Figure 7 ijms-23-12636-f007:**
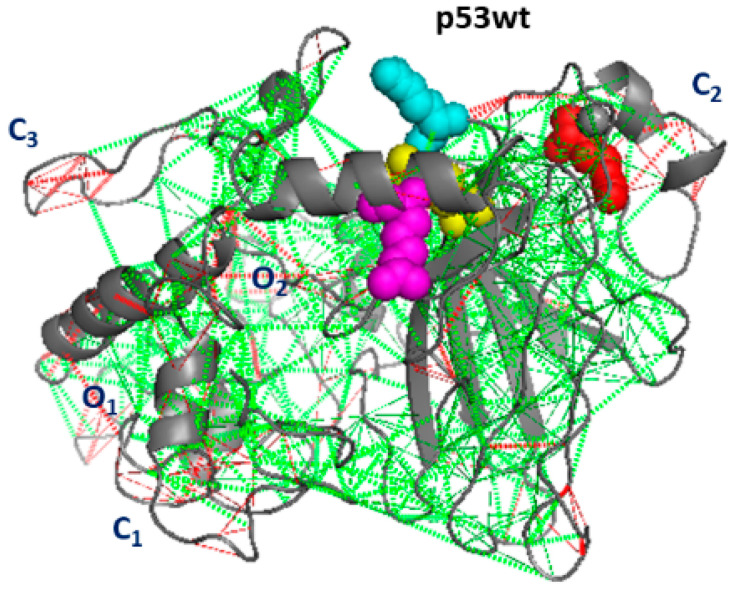
Frustration snapshot of p53wt.The level of frustration is represented by the colored scale: green represents minimal frustration and red high frustration. The residues submitted to punctual mutations and the Zn ion are represented as spheres.

**Figure 8 ijms-23-12636-f008:**
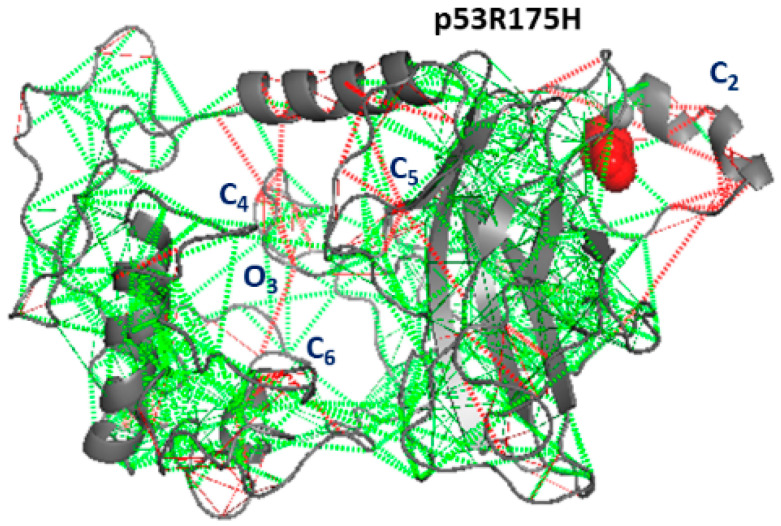
Frustration analysis of p53R175H. The level of frustration is represented by the colored scale: green represents minimal frustration and red high frustration. The mutated residue (175H, red) and the Zn ion (grey) are represented as spheres.

**Figure 9 ijms-23-12636-f009:**
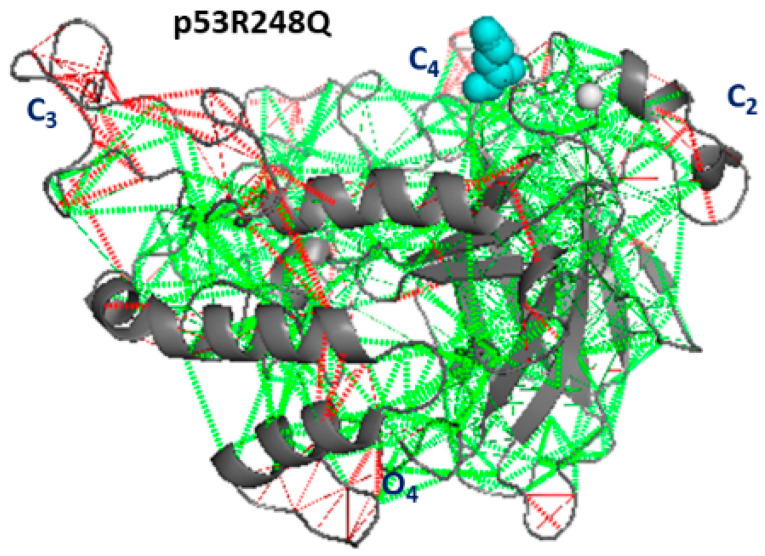
Frustration analysis of p53R248H. The level of frustration is represented by the colored scale: green represents minimal frustration and red high frustration. The mutated residue (248H, cyan) and the Zn ion (grey) are represented as spheres.

**Figure 10 ijms-23-12636-f010:**
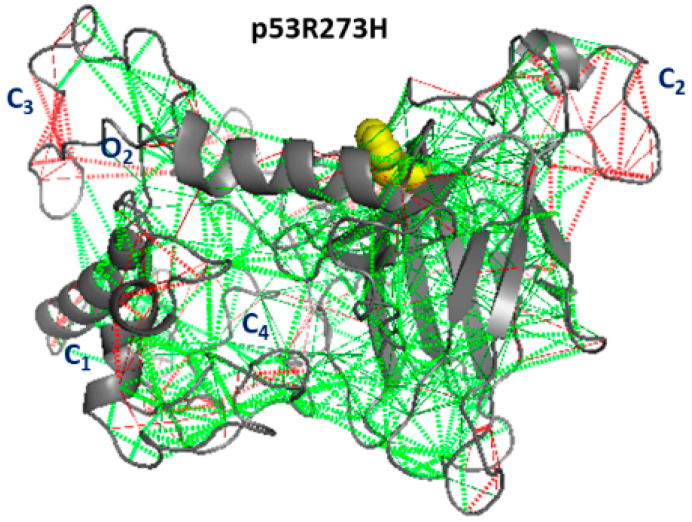
Frustration analysis of p53R273H. The level of frustration is represented by the coloured scale: green represents minimal frustration and red high frustration. The mutated residue (273H, yellow) and the Zn ion are represented as spheres.

**Figure 11 ijms-23-12636-f011:**
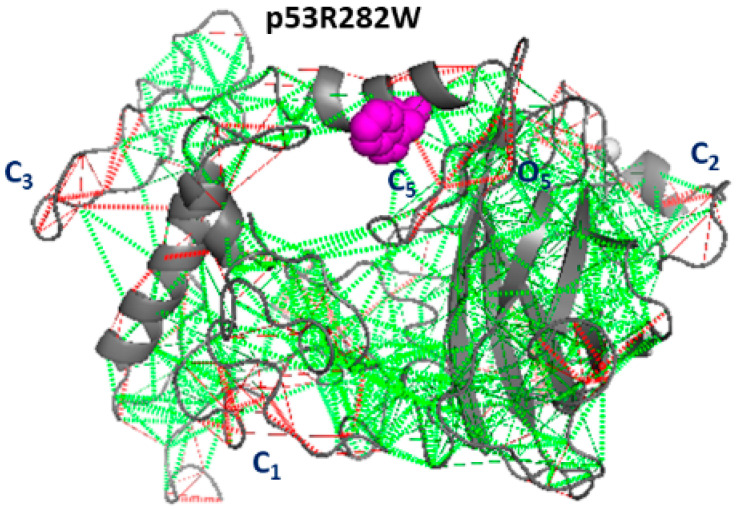
Frustration analysis of p53R282W. The level of frustration is represented by the colored scale: green represents minimal frustration and red high frustration. The mutated residue (282W, magenta) and the Zn ion (grey) are represented as spheres.

**Table 1 ijms-23-12636-t001:** General properties of p53wt and the four p53 mutants. RMSD, SASA, and Rg were evaluated from the 50–150 ns interval of the runs, while the percentage of the secondary structure was evaluated from the average structure from the last 10 ns (141–150 ns) of the runs.

System	RMSD (nm)	SASA (nm^2^)	R_g_ (nm)	Secondary Structure (%)
**p53wt**	0.50 ± 0.06	216 ± 6	2.22 ± 0.05	64 ± 6
**p53R175H**	0.53 ± 0.06	215 ± 6	2.26 ± 0.04	63 ± 8
**p53R248Q**	0.71 ± 0.05	220 ± 3	2.26 ± 0.04	67 ± 9
**p53R273H**	0.46 ± 0.04	214 ± 4	2.24 ± 0.05	64 ± 8
**p53R282W**	0.54 ± 0.05	217 ± 5	2.25 ± 0.05	65 ± 7

**Table 2 ijms-23-12636-t002:** ED and frustration properties of p53wt and of p53 mutants. ED parameters were derived from the 50–150 ns interval. Frustration index was evaluated from the average structure from the las 10 ns (141–150 ns).

System	Sum of First Ten/Sum of All (%)	Width PC1 (nm)	Width PC2 (nm)	Frustration Index %
**p53wt**	67 ± 8	35 ± 2	22 ± 4	7.3 ± 0.6
**p53R175H**	62 ± 3	31 ± 2	24 ± 1	8.6 ± 0.7
**p53R248Q**	71 ± 7	49 ± 7	32 ± 4	8.9 ± 0.9
**p53R273H**	66 ± 8	38 ± 3	26 ± 3	8.2 ± 0.6
**p53R282W**	62 ± 9	31 ± 2	24 ± 2	7.8 ± 0.9

## Data Availability

Not applicable.

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
