# Peer review of "Conformational Heterogeneity and Frustration of the Tumor Suppressor p53 as Tuned by Punctual Mutations"

_ijms, 2022, doi:10.3390/ijms232012636_

Round 1

Reviewer 1 Report

In this study, the author performed 5*150 ns MD simulations on wild-type tumor suppressor p53 and its four mutated forms (p53R175H, p53R248Q, p53R273H, p53R282W), based on a full-length, homology-modeled structure. Using essential dynamics and frustration analyses, the author characterized their collective motions and the degeneracy of their energy landscapes. They reported that p53 is characterized by wide collective motions and its energy landscape exhibits high frustration level in the ligand-binding regions. Also, those mutations help modulate the collective motions and frustration level, thereby leading to functional regulations. It’s an interesting study overall. The authors thoroughly analyzed their simulation data and presented their results in well organized sections in the manuscript. However, there are several major points that need to be addressed. In addition, the writing of the manuscript needs to be improved, as there are some grammar errors. I’d recommend the authors to address those points, before considering the publication of this manuscript in IJMS. 

Specific points:

  1. How many residues are missing in the crystal structure of p53 with DNA (PDB ID: 1TUP)? Are they mostly in N- and C-terminal regions, which are intrinsically disordered? Please provide those details in the method sections for evaluating the model quality. In addition, please provide additional method details (system size and box dimensions etc). 

  1. “Here, we are interested to closely address such a conformational heterogeneity in the same temporal window and in the whole p53 molecule.” Is 100 ns sufficient to capture the conformational ensemble of p53? Given that the structural model is built from a homology model and there are significant disordered regions, I doubt that’s sufficient to capture the conformational heterogeneity. The author may consider running longer MD simulations and show that the conformational landscape from the 100 ns simulations do not change significantly with longer simulations. That’s helpful for showing the convergence of simulations and analyses. Given the size of the system, it should not be expensive to do additional calculations.

  1. “allows to extract collective modes from the conformational space explored by a biomolecule during the MD runs” - Please provide a brief explanation of the physical meaning of collective mode of a biomolecule. 

  1. “The first 30 eigenvalues, 149 extracted from the covariance matrix of the atomic fluctuations of representative runs, and 150 ranked in a decremental order, are shown in Figure 3A.” Please provide a brief explanation of the physical meanings of eigenvalue and eigenvector from the ED method. 

  1. It would be helpful if the author can utilize quantitative metrics to measure the collective modes for random coils and alpha-helices vs. beta-sheets, when the wild-type and the mutated forms are compared in Figure 4 and Figure 5. In the current manuscript, very vague terms are used to compare the difference, such as “lesser extent”. 

  1. When discussing total and per-residue frustration index, please also provide a brief explanation on how to interpret this index. That will be helpful for the readers.

  1. Unfortunately, it’s very difficult to compare and interpret the visualization of frustration analysis in Figure 6-10A. Would it help if only plotting all residue pairs with high frustration, or plotting the residue pairs with high/low frustration separately? 

  1. For the residue-residue frustration index plots (Figure 6-10B), the author should clearly explain how they identify the clusters as labeled and discussed in the manuscript. It appears to the reviewer that they were just chosen randomly. They are clearly not the only clusters that can be found in the plots. Some of the identified clusters do not appear to be significant either. Please provide captions for this plot in Figure 6-10B.

  1. The author only wrote 1 short paragraph discussing the relationship between collective motions and high frustration by referring to the wild-type p53 analyses in Figure 4 and 6. What about other mutants? Do you see similar patterns as observed in wild type p53? The author may consider elaborating the discussion here and use additional figures and data to support their claims. 

  1. How can those findings from collective mode and frustration analyses be utilized to explain the functional effects of those mutations? Are there any previous structural studies on those mutated p53, that can be referenced and compared to this study?

  1. “These results can help to elucidate the intricate relationship between the structural and dynamical of p53 and its functionality. Furthermore, they also provide some clues on how to modify the functionality of a biomolecule through a punctual mutation leading to an appropriate modulation of the energy landscape.” The reviewer still finds it difficult to see the impact of those theoretical analyses in understanding the structure-function relationship of p53. 

  1. Please carefully revise the manuscript to correct small grammar errors in the manuscript.

Author Response

In this study, the author performed 5*150 ns MD simulations on wild-type tumor suppressor p53 and its four mutated forms (p53R175H, p53R248Q, p53R273H, p53R282W), based on a full-length, homology-modeled structure. Using essential dynamics and frustration analyses, the author characterized their collective motions and the degeneracy of their energy landscapes. They reported that p53 is characterized by wide collective motions and its energy landscape exhibits high frustration level in the ligand-binding regions. Also, those mutations help modulate the collective motions and frustration level, thereby leading to functional regulations. It’s an interesting study overall. The authors thoroughly analyzed their simulation data and presented their results in well organized sections in the manuscript. However, there are several major points that need to be addressed. In addition, the writing of the manuscript needs to be improved, as there are some grammar errors. I’d recommend the authors to address those points, before considering the publication of this manuscript in IJMS. 

I would like to thank the Reviewer for his/her appreciation of the work.

 Specific points:

  1. How many residues are missing in the crystal structure of p53 with DNA (PDB ID: 1TUP)? Are they mostly in N- and C-terminal regions, which are intrinsically disordered? Please provide those details in the method sections for evaluating the model quality. In addition, please provide additional method details (system size and box dimensions etc). 

The required details have been added to the Methods Section. In particular, the number of missing residues, and the regions to which they belong have been now mentioned in Section 3.1 (Pag.14 Lines 431-435). Further details on the followed simulation procedures have been explicitly added in Section 3.2 (Pag.13 Lines 452-457).

  1. Here, we are interested to closely address such a conformational heterogeneity in the same temporal window and in the whole p53 molecule.” Is 100 ns sufficient to capture the conformational ensemble of p53? Given that the structural model is built from a homology model and there are significant disordered regions, I doubt that’s sufficient to capture the conformational heterogeneity. The author may consider running longer MD simulations and show that the conformational landscape from the 100 ns simulations do not change significantly with longer simulations. That’s helpful for showing the convergence of simulations and analyses. Given the size of the system, it should not be expensive to do additional calculations.

We agree with the Reviewer that 100 ns could be not long enough to catch the whole structural heterogeneity of the p53 system. However, our aim is to investigate the structural heterogeneity of p53 in the 100 ns temporal window, in comparison with the previous, related results from time-resolved fluorescence data. Such an aspect has been now stressed in the Introduction (Pag.2 Lines 64-65). Furthermore, to support the reliability of the analyzed temporal window to reach the convergence, the temporal evolution of the RMSD has been shown together with a related comment (see new Figure 1B and Pag.3 Lines 117-127).

  1. “allows to extract collective modes from the conformational space explored by a biomolecule during the MD runs” - Please provide a brief explanation of the physical meaning of collective mode of a biomolecule. 

A brief explanation about the physical meaning of collective modes in biomolecules has been added in the Introduction of the manuscript (Pag.2 Lines 76-78).

  1. The first 30 eigenvalues, 149 extracted from the covariance matrix of the atomic fluctuations of representative runs, and 150 ranked in a decremental order, are shown in Figure 3A.” Please provide a brief explanation of the physical meanings of eigenvalue and eigenvector from the ED method. 

Additional information about the physical meaning of eigenvalues and eigenvectors in the framework of the ED method has been added to the manuscript (Pag.5 Lines 181-185)

  1. It would be helpful if the author can utilize quantitative metrics to measure the collective modes for random coils and alpha-helices vs. beta-sheets, when the wild-type and the mutated forms are compared in Figure 4 and Figure 5. In the current manuscript, very vague terms are used to compare the difference, such as “lesser extent”.

The differences between the two extreme projections of the eigenvectors have been calculated and the regions characterized by differences higher than 0.6 nm have been marked in both Figures 4 and 5. The discussion about the related results has been modified, accordingly (Pag.6 Lines 240-241, Pag.7 Lines 256-259 and new Figures 4 and 5).

  1. When discussing total and per-residue frustration index, please also provide a brief explanation on how to interpret this index. That will be helpful for the readers.

A more detailed description of the frustration index has been added to the manuscript (Pag.8 Lines 302-303, Pag.9 304-308).

  1. Unfortunately, it’s very difficult to compare and interpret the visualization of frustration analysis in Figure 6-10A. Would it help if only plotting all residue pairs with high frustration, or plotting the residue pairs with high/low frustration separately? 
  2. For the residue-residue frustration index plots (Figure 6-10B), the author should clearly explain how they identify the clusters as labeled and discussed in the manuscript. It appears to the reviewer that they were just chosen randomly. They are clearly not the only clusters that can be found in the plots. Some of the identified clusters do not appear to be significant either. Please provide captions for this plot in Figure 6-10B.

The frustration analysis, has been deeply revised to better put into evidence the results obtained for the analyzed systems. In particular, the frustration map has been shown only for p53wt, and the snapshots of all the p53 molecules have been enlarged for a better visualization of the changes. Additionally, the identification of the most relevant clusters has been clearly specified and the related effects more extensively described (see new Figures 6-11 and the revised Section 2.3).

  1. The author only wrote 1 short paragraph discussing the relationship between collective motions and high frustration by referring to the wild-type p53 analyses in Figure 4 and 6. What about other mutants? Do you see similar patterns as observed in wild type p53? The author may consider elaborating the discussion here and use additional figures and data to support their claims. 
  2. How can those findings from collective mode and frustration analyses be utilized to explain the functional effects of those mutations? Are there any previous structural studies on those mutated p53, that can be referenced and compared to this study?

The discussion about the relationship between the establishment of collective motions and the observation of high frustration, has been further developed also with the help of a direct comparison among new Figures 4 and 5, in which collective motions have been evidenced (as mentioned in the answer to Point 5), (see the new Figures 8-11 and the revised Section 2.3)

11.These results can help to elucidate the intricate relationship between the structural and dynamical of p53 and its functionality. Furthermore, they also provide some clues on how to modify the functionality of a biomolecule through a punctual mutation leading to an appropriate modulation of the energy landscape.” The reviewer still finds it difficult to see the impact of those theoretical analyses in understanding the structure-function relationship of p53. 

We agree with the Reviewer that the information extracted from collective motions and frustration could be not directly used to improve p53 functionality, however, we believe that they might improve the knowledge on p53 and its network. Therefore, to address the Reviewer’s observation, the related sentence in the Conclusions has been revised (Pag.15 Lines 523-527).

  1. Please carefully revise the manuscript to correct small grammar errors in the manuscript.

The manuscript has been carefully revised.

Reviewer 2 Report

Lines 208-220 = For describing results for Figure 5, please use Table 2 as reference as Table is more informative than the figure in this case.

Authors have nice results about frustration in wtp53 and its mutants. Authors could comment on possible experimental methods to confirm the structural/conformational differences in wt p53 and its mutants. For e.g Hydrogen deuterium exchange mass spectrometry and Small-angle X-ray scattering could be used. Following manuscripts can provide reference for these techniques.

https://www.pnas.org/doi/abs/10.1073/pnas.2019571118

https://link.springer.com/article/10.1007/s00249-015-1079-9

Author Response

Lines 208-220 = For describing results for Figure 5, please use Table 2 as reference as Table is more informative than the figure in this case.

The description of results from Figure 5 has been done also by using data from Table 2. On the other hand, we mention that Figure 4 and Figure 5 have been modified to better put into evidence the result discussion (see new Figures 4 and 5 and Pag.8  Lines 278-280).

Authors have nice results about frustration in wtp53 and its mutants. Authors could comment on possible experimental methods to confirm the structural/conformational differences in wt p53 and its mutants. For e.g Hydrogen deuterium exchange mass spectrometry and Small-angle X-ray scattering could be used. Following manuscripts can provide reference for these techniques. https://www.pnas.org/doi/abs/10.1073/pnas.2019571118; https://link.springer.com/article/10.1007/s00249-015-1079-9;

A comment about possible experimental methods to investigate conformational differences between p53 and its mutants has been added together with some newly added references (new refs. [39-42] and Pag.4 Lines 169-173).

Round 2

Reviewer 1 Report

The authors have addressed the reviewer's comments and concerns from the first round of review. The manuscript has been significantly improved. It can be accepted for publication in IJMS in the present form.